# *SLC5A1* Variants in Turkish Patients with Congenital Glucose-Galactose Malabsorption

**DOI:** 10.3390/genes14071359

**Published:** 2023-06-27

**Authors:** Ferda Ö. Hoşnut, Andreas R. Janecke, Gülseren Şahin, Georg F. Vogel, Naz G. Lafcı, Paul Bichler, Thomas Müller, Lukas A. Huber, Taras Valovka, Aysel Ü. Aksu

**Affiliations:** 1Department of Pediatric Gastroenterology, Hepatology and Nutrition, Dr. Sami Ulus Maternity and Child Health and Diseases Training and Research Hospital, University of Health Sciences, 06080 Ankara, Turkey; 2Department of Pediatrics I, Medical University of Innsbruck, 6020 Innsbruck, Austriataras.valovka@i-med.ac.at (T.V.); 3Institute of Human Genetics, Medical University of Innsbruck, 6020 Innsbruck, Austria; 4Institute of Cell Biology, Biocenter, Medical University of Innsbruck, 6020 Innsbruck, Austria; 5Department of Medical Genetics, Faculty of Medicine, Hacettepe University, 06230 Ankara, Turkey; 6Department of Medical Genetics, Dr. Sami Ulus Maternity and Child Health and Diseases Training and Research Hospital, University of Health Sciences, 06080 Ankara, Turkey; 7Department of Pediatric Gastroenterology, Hepatology and Nutrition, Ankara Bilkent Hospital, University of Health Sciences, 06800 Ankara, Turkey; ayselun@gmail.com

**Keywords:** glucose-galactose malabsorption, SGLT1, missense mutation, transmembrane domain, glycosylation, endoplasmic reticulum

## Abstract

Congenital glucose-galactose malabsorption is a rare autosomal recessive disorder caused by mutations in *SLC5A1* encoding the apical sodium/glucose cotransporter SGLT1. We present clinical and molecular data from eleven affected individuals with congenital glucose-galactose malabsorption from four unrelated, consanguineous Turkish families. Early recognition and timely management by eliminating glucose and galactose from the diet are fundamental for affected individuals to survive and develop normally. We identified novel *SLC5A1* missense variants, p.Gly43Arg and p.Ala92Val, which were linked to disease in two families. Stable expression in CaCo-2 cells showed that the p.Ala92Val variant did not reach the plasma membrane, but was retained in the endoplasmic reticulum. The p.Gly43Arg variant, however, displayed processing and plasma membrane localization comparable to wild-type SGLT1. Glycine-43 displays nearly invariant conservation in the relevant structural family of cotransporters and exchangers, and localizes to SGLT1 transmembrane domain TM0. p.Gly43Arg represents the first disease-associated variant in TM0; however, the role of TM0 in the SGLT1 function has not been established. In summary, we are expanding the mutational spectrum of this rare disorder.

## 1. Introduction

Congenital glucose-galactose malabsorption (cGGM, OMIM #606824) is a rare, life-threatening autosomal recessive disorder caused by defects in the active transport of glucose and galactose across the intestinal brush border [1]. cGGM is characterized by frequent, abundant osmotic diarrhea, severe hypernatremic dehydration, and metabolic acidosis, beginning in the neonatal period due to the accumulation of unabsorbed sugars in the intestinal lumen [2,3]. Biallelic *SLC5A1* (OMIM #182380) variants underlying cGGM were first reported in 1991 [4]. *SLC5A1* is located on 22q12.3 and encodes one of the members of the sodium-dependent glucose transporter (SGLT) family, enabling the transfer of glucose and galactose coupled to transmembrane sodium transport [1,5]. The protein encoded from this gene contains 14 transmembrane domains (TM0-TM13) [6] and is mainly expressed in the intestine and the proximal tubules of the kidney.

Since the first description in 1962, more than 300 affected individuals of diverse ethnic origins have been reported worldwide, most of whom are born to consanguineous unions [2,7]. Molecular studies have identified 69 *SLC5A1* variants, as detailed by the Human Gene Mutation Database Professional 2021.4 (http://www.hgmd.cf.ac.uk/ac/gene.php?gene=SLC5A1, accessed on 8 November 2022), of which the majority are missense and nonsense, with the remainder consisting of splicing variants and small insertions and deletions. Defects of this transporter lead to diarrhea that ceases with fasting or withdrawal of the offending sugars from the diet, but resumes immediately with oral feeding of diets containing lactose, glucose, or galactose. A fructose-based formula that does not contain glucose or galactose provides rapid recovery from diarrhea, since fructose absorption is unaffected as fructose is taken up by another transporter (GLUT5) [8,9]. 

There are two case reports about affected individuals with cGGM of Turkish origin in the literature [2,10], and here we present a cohort of eleven affected individuals from four unrelated families from a rural area of Turkey.

## 2. Materials and Methods

### 2.1. Recruitment of cGGM Patients 

This retrospective observational study was conducted at a tertiary hospital, and the study protocol was approved by the Institutional Ethics Committee of Ankara City Hospital (protocol number: E1-20-1062). All of the individuals gave full informed consent to participate in this study. Between 2011 and 2019, we retrospectively identified all affected individuals with the clinical diagnosis of cGGM based on symptoms and disease course consistent with cGGM and responsive to the appropriate fructose-based diet followed at Dr. Sami Ulus Maternity and Child Health and Diseases Training and Research Hospital. Their complete medical history was obtained and reviewed to determine the age at presentation, gender, clinical symptoms (presence and severity of diarrhea, dehydration, nephrocalcinosis, and growth), genetic testing results, treatment, and overall outcome for each affected individual. Details regarding parental consanguinity and similar disease history in relatives were obtained during the interviews with the families. 

### 2.2. Identification of SLC5A1 Mutations 

Molecular studies were performed in Austria, at the Department of Pediatrics, and at the Institute of Human Genetics at the Medical University of Innsbruck with approval by the Institutional Ethics Committee (protocol number: AN2016-0029). Genomic DNA was isolated from peripheral blood leukocytes by standard procedures from the affected individuals, their parents, and siblings. Molecular studies were conducted in the index individuals of each family, either by Sanger sequencing of the complete coding region and all flanking splice-sites of *SLC5A1* (family 3) or whole exome sequencing (WES) in all other families, and followed by variant filtering in genes associated with known entities of congenital diarrhea. The whole exome target enrichment was performed from the genomic DNA of the index individuals in Families 1, 2, and 4 using the Agilent SureSelect V6 exome Kit (Agilent Technologies, Santa Clara, CA, USA). A total of 150 bp paired-end reads were generated using an Illumina HiSeq 2500 platform (Illumina, San Diego, CA, USA). Variant filtering was performed according to the autosomal recessive mode of inheritance and excluding common variants with minor allele frequency (MAF > 0.01) in population databases (dbSNP, ExAC, and gnomAD). Variants in known OMIM-listed disease genes implicated in congenital diarrheas were identified. All identified variants in the index individuals were confirmed, and segregation analyses were completed by Sanger sequencing of the identified *SLC5A1* variants in all available family members. The *SLC5A1* variant designation is based on the NCBI reference sequence for transcript *NM_000343.4*. For the in silico prediction of the identified variants, CADD (Combined Annotation Dependent Depletion; https://cadd.gs.washington.edu/, accessed on 8 November 2022), Mutation Taster (http://www.mutationtaster.org/, accessed on 8 November 2022), PolyPhen-2 (prediction of functional effects of human non-synonymous SNPs; http://genetics.bwh.harvard.edu/pph2/, accessed on 8 November 2022), and SIFT (Sorting Intolerant From Tolerant; https://sift.bii.a-star.edu.sg/, accessed on 8 November 2022) were used to evaluate the pathogenicity of the identified variants on the biological function of the protein. Finally, evolutionary conservation of the substituted amino acids due to the missense variants in *SLC5A1* was analyzed using Clustal Omega (http://www.ebi.ac.uk/Tools/msa/clustalo/, accessed on 8 November 2022), a multiple sequence alignment program. Whole-genome sequencing was performed in patient 4 in order to display potentially pathogenic intronic *SLC5A1* variants that might underlie the phenotype using Illumina short-read sequencing (Eurofins, Germany).

### 2.3. Antibodies and Reagents 

Primary ani-HA (16B12) antibody was used for Western blotting (WB), at a dilution of 1:1000, MMS-101R; Covance, and for immunofluorescence microscopy (IF) at a dilution of 1:500; anti-β-Actin (WB, 1:2000; #4967; Cell Signaling Technology Europe B.V., Frankfurt am Main, Germany). Secondary horseradish peroxidase-conjugated goat anti-mouse and goat anti-rabbit (1:4000; Sigma-Aldrich (Merck, Darmstadt, Germany)) were used for immunoblotting. Actin filaments were labeled with phalloidin Alexa Fluor 568 (1:500; A12380; Invitrogen, Life Technologies Austria, Branch Office Vienna, Austria); anti-Calnexin (C5C9) antibodies were used to visualize endoplasmic reticulum (ER) (1:200; #2679; Cell Signaling Technology). Secondary Alexa Fluor-conjugated (Alexa Fluor 488 and 568) goat anti-mouse (1:1000; Invitrogen), goat anti-rabbit (1:1000; Invitrogen), and Hoechst 33342 (1:10,000; B2261; Sigma-Aldrich) were used for IF labeling. N-glycosidase F (PNGase-F) and endoglycosidase H (Endo H) were purchased from New England Biolabs.

### 2.4. Construction of cDNA Clones

The pRRL CMV GFP Sin-18 plasmid [11] was used to generate the *SLC5A1* expressing viral construct. For this, the human cytomegalovirus (CMV) promoter was replaced with the elongation factor 1a short (EFS-NS) promoter, and the N-terminal HA-tagged *SLC5A1* cDNA was cloned into the BamHI and SalI sites of the resulting plasmid. A cap-independent translation enhancer (CITE) fused to the puromycin resistance *pac* gene and the woodchuck hepatitis virus post-transcriptional regulatory element (WPRE) were introduced downstream of the *SLC5A1* coding sequence. *SLC5A1* mutations were generated by site-directed mutagenesis PCR using the QuikChange protocol (Stratagene, Sigma-Aldrich). Sequences of single-strand oligonucleotide pairs used for mutagenesis are available from the authors. All generated constructs were sequence verified.

### 2.5. Cell Culture

The human intestinal carcinoma cell line CaCo-2 and human embryonic kidney HEK 293T cells were cultured in Dulbecco’s Modified Eagle Medium (DMEM; Sigma–Aldrich) supplemented with 10% FBS, 100 U/mL penicillin, and 100 μg/mL streptomycin in a humidified atmosphere with 5% CO_2_ at 37 °C. To establish polarized epithelial cell monolayers, CaCo-2 cells were seeded at a high density and cultured for 2–3 weeks with a regular change (every 3 days) of the culture medium.

### 2.6. Stable Expression of WT and Mutant SLC5A1 Alleles in CaCo-2 Cells 

Lentiviral particles were produced by transfecting HEK 293T cells with the pRRL- HA-SLC5A1 transfer plasmid, packaging psPAX2, and VSV-G envelopepMD.G plasmids as described previously [12]. The viral supernatants were collected, concentrated with Retro-X Concentrator (Clontech, Takara Bio USA Inc., Mountain View, CA, USA), and used to transduce CaCo-2 cells in the presence of polybrene (4 μg/mL) (Sigma–Aldrich, Merck, Darmstadt, Germany). Positively transduced cells were selected with puromycin (Gibco, Thermo Fisher Scientific, Loughborough, UK) at a concentration of 10 µg/mL. 

### 2.7. Transient Expression of WT and Mutant SLC5A1 Alleles in HEK 293T Cells 

For transient expression, HEK 293T cells were transfected with the pRRL-HA-SLC5A1 constructs using the calcium phosphate co-precipitation method. Five hours after transfection, the culture medium was replaced with a fresh one, and the cells were then grown for 24–36 h prior to analysis.

### 2.8. Detergent Fractionation and Immunoblotting 

For analysis of detergent-soluble and -insoluble fractions, HEK 293T cells and CaCo-2 cell monolayers were washed with 1× PBS and lysed in ice-cold Nonidet P40 lysis buffer (40 mM Tris-HCl pH 8.0, 100 mM NaCl, 0.5% (*v*/*v*) Nonidet-P40, 10 mM β-glycerophosphate, 10 mM NaF, 1 mM EDTA, 1 mM PMSF, 1 μg/mL pepstatin, 1 μg/mL aprotinin, and 1 μg/mL leupeptin) for 10 min on ice. Cell extracts were centrifuged at 10,000× *g* and 4 °C for 20 min. The supernatants, representing the detergent-soluble fraction, were collected, and the detergent-resistant pellets were washed once with the lysis buffer, solubilized in 2× SDS-PAGE sample buffer (125 mM Tris-HCl pH 6.8, 20% (*v*/*v*) glycerol, 2% (*v*/*v*) SDS, 100 mM DTT, 0.005% bromophenol blue), sonicated, and boiled at 95 °C for 5 min. To normalize the samples, protein concentration was determined in supernatants using Coomassie Plus Protein Assay Reagent (Thermo Scientific, Waltham, MA, USA). Protein extracts in sample buffer were separated by SDS–PAGE, transferred to a nitrocellulose membrane (Amersham Protran Premium 0.2 µm NC), blocked with 5% milk protein in TBST (20 mM Tris-HCl pH 7.6, 137 mM NaCl, 0.05% (*v*/*v*) Tween), and immunoprobed with the antibodies indicated according to manufacturer’s recommendations. Enhanced chemiluminescence detection (Biozym Scientific, Oldendorf, Germany) was used in immunoblot assays.

### 2.9. Deglycosylation of Protein Extracts

Fifty micrograms of detergent-soluble protein extracts were treated with 250 U of N-glycosidase F (PNGase-F) or 250 U of endoglycosidase H (Endo H) in 1× GlycoBuffer2 or GlycoBuffer3 (New England Biolabs, Frankfurt, Germany) at 37 °C for 1 h, respectively. The reactions were stopped by adding 5× SDS-PAGE sample buffer and incubating at 37 °C for 15min. Samples were stored at −20 °C and analyzed by immunoblotting.

### 2.10. Immunofluorescence and Imaging

Immunofluorescent labeling of CaCo-2 cells grown to confluence on coverslips was performed as described previously (Vogel et al., 2015). Confocal images were taken with a confocal fluorescence microscope (SP5; Leica, Vienna, Austria) using a glycerol 63× lens with a numerical aperture of 1.3 (Leica) at room temperature and mounted in Mowiol. The recording software used was LASAF 2.7.3 (Leica). Images were deconvolved and adjusted for brightness and contrast with Image J software version 1.53t (https://imagej.net/ij/index.html). HEK 293T cells were analyzed with an epifluorescent Axio Imager M1 microscope (Carl Zeiss, Oberkochen, Germany) equipped with a charge-coupled device camera (SPOT Xplorer; Visitron Systems, Puchheim, Germany) and recorded with VisView 2.0.3 (Visitron Systems). 

### 2.11. Human SGLT1 Protein Structure Modeling 

To model the human SGLT1 p.Ala92Val variant and calculate the vibrational entropy changes between wild-type and mutant, the DynaMut web server [13,14] and the cryoEM structure of hSGLT1 (PDB: 7SLA) [6] were used. Non-covalent molecular interactions in the model were calculated using Arpeggio [14]. The resulting change in free energy was predicted using the Elastic Network Contact Model (ENCoM) method [15].

## 3. Results

### 3.1. Clinical Findings

Forty-two individuals, of whom eleven were affected with cGGM from four families, were included in this study. Six (54.5%) of the affected individuals were female, and five (45.5%) were male. All sets of parents were first-degree cousins. The second and third families are composed of two and three different branches, respectively. All families were from the same city, where people of Arab and Kurdish origin lived densely.

Birth weights of all newborns were within normal limits, and all were born after an uneventful full-term pregnancy. The disease initially manifested as persistent diarrhea, which occurred in all infants during the first two weeks after birth, after they had been breastfed or bottle-fed, and at admission, affected individuals presented with moderate or severe hypernatremic dehydration and metabolic acidosis. The clinical and laboratory characteristics of the eight affected individuals are detailed in Table 1. Renal failure was not documented in any of the individuals, whereas nephrocalcinosis was detected by renal ultrasound in three affected individuals. The diarrhea of the affected individuals could only be controlled with the formula (Galactomin-19^®^, SHS International, Liverpool, UK) containing fructose within a few days. Three additional children had been hospitalized due to diarrhea that started in the neonatal period and had recovered with the formula containing fructose, indicating cGGM as the diagnosis, which was supported by *SLC5A1* mutation analysis (see below); however, detailed clinical and laboratory findings were not available. Part of the medical history of family four has been recently reported [16]. Of the eight affected individuals who were followed up in our center, the heights of four children trended below -2 standard deviation score (SDS), one of whom also had a weight below -2 SDS. 

Of note, two siblings in family one, two siblings in the first branch of family two, one child in the first branch of family three, and three siblings in the second branch of the third family had died due to diarrhea in the neonatal period. Eleven affected individuals in our study began to tolerate foods containing carbohydrates as they grew older. There was no significant complaint of diarrhea in these individuals unless there was excessive carbohydrate consumption. A possible diagnosis of cGGM was suspected in each family based on the clinical history and laboratory findings.

### 3.2. Identification of Pathogenic and Likely Pathogenic SLC5A1 Variants

Molecular studies revealed rare *SLC5A1* variants in all families (Figure 1, Table 2). The identified *SLC5A1* variants were considered pathogenic for the following reasons:(1)p.Arg267* has a very low allele frequency, and p.Ala92Val and p.Gly43Arg are absent from the population database gnomAD, discounting them to represent frequent sequence variants;(2)All three variants segregate with disease in each family;(3)p.Arg267* causes an early, premature stop codon, predicting abrogation of protein production; this variant has been reported once, in a homozygous state, in a Turkish patient with cGGM [17];(4)The novel *SLC5A1* variant c.275C>T (p.Ala92Val) was found in the index patient from family two in a homozygous state, and it resided within a large homozygous region of 23 Mb on chromosome 22q12.3, indicating inheritance by identity by descent and providing evidence for its pathogenicity. This novel variant replaces a highly conserved amino acid (Figure 2), and multiple lines of in silico prediction analyses supported its pathogenicity (PolyPhen2 score: 1.00, CADD score: 19, 76, SIFT score: 0.02);(5)The novel *SLC5A1* variant c.127G>A (p.Gly43Arg) was found in the index in family four in the homozygous state, and it resided within a large homozygous region of 17 Mb on chromosome 22q12.3, indicating inheritance by identity by descent and providing evidence for its pathogenicity. This novel variant replaces a nearly invariantly conserved amino acid (Figure 2), and multiple lines of in silico prediction analyses supported its pathogenicity (PolyPhen2 score: 0.998, CADD score: 22, SIFT score: 0); whole-genome sequencing was performed, and did not detect potentially disease-causing *SLC5A1* variants in cis;(6)Moreover, WES did not identify other pathogenic variants related to this disease in other genes in families one, two, and four.

### 3.3. Modeling SLC5A1 Variant p.Ala92Val Provides Evidence for Its Pathogenicity

Alanine-92 resides at the extracellular end of transmembrane (TM) helix 1b with the hydrophobic sidechain exposed outside the TM substrate-translocation cavity, suggesting a structural role rather than direct involvement in the gating or binding of sugars. Mutating alanine-92 to valine had a mild effect on the stability of SGLT1 (change in free energy of 0.273 kcal/mol) and caused an overall decrease in molecular flexibility (ΔΔS_Vib_ = −0.342 kcal.mol^−1^·K^−1^) (Figure 3A). Compared with alanine-92, valine-92 forms additional hydrogen bonds with the carbonyl groups of Ala88 and Gly89 of TM1 and the sidechain hydroxyl group of Tyr236 of extracellular loop 3 (EL3). The substitution of valine also enables strong ionic and hydrophobic interactions with Gly332 and Arg336 of the TM7 helix, respectively (Figure 3B). 

### 3.4. Transient and Stable Expression of Novel SLC5A1 Variants in HEK293T and CaCo-2 Cells

To demonstrate the potential effects of novel variants on the sorting and trafficking of SGLT1, CaCo-2 cells were stably transduced with lentiviruses encoding for HA-SGLT1 wild-type, p.Gly43Arg or p.Ala92Val. The expression of ectopic wild-type SGLT1 was detected by immunoblotting in detergent-soluble (S) fraction as ~60 kDa and ~70 kDa bands and in detergent-resistant (P) fraction, generally enriched in membrane rafts and cytoskeleton proteins, as a ~70 kDa band (Figure 4A). Treatment of lysates with N-glycosidase F (PNGase-F) and endoglycosidase H (Endo H) revealed that ~70 kDa bands were Endo H resistant complex-glycosylated forms of SGLT1, whereas the ~60 kDa specific band represented core-glycosylated proteins (Figure 4B). While the p.Gly43Arg variant behaved comparable to wild-type SGLT1, the p.Ala92Val variant was missing complex glycosylation and membrane enrichment (Figure 4A,B).

These data prompted us to compare the localization and trafficking of SGLT1 variants to the apical plasma membrane in polarized CaCo-2 cells. Consistent with previous reports [18], the wild-type SGLT1 protein was detected as sparse punctuated structures in the apical membrane but was not strictly apical, and a substantial portion of the protein localized in a dispersed manner throughout the cytoplasm (Figure 4C). A similar pattern of cellular distribution was observed for the p.Gly43Arg mutant, suggesting that this mutation did not compromise the localization of the transporter to the apical surface of enterocytes. In contrast, the p.Ala92Val variant was mostly depleted from the apical brush border and only resided intracellularly with prominent basal staining (Figure 4C).

To assess the processing and trafficking of newly translated SGLT1 variants, we analyzed the glycosylation and cellular distribution of SGLT1 in transiently transfected human embryonic kidney HEK 293T cells (Figure 5).

Similarly to CaCo-2 cells, substantial amounts of SGLT1 wild-type and p.Gly43Arg mutant were expressed as Endo H resistant forms, whereas only a minor portion of p.Ala92Val protein was complex-glycosylated in transiently transfected HEK 293T cells (Figure 5A). Transmembrane proteins, such as SGLT1, are normally core-glycosylated in the endoplasmic reticulum and require translocation to the cis and trans-Golgi networks for complex N-glycan processing. Because the SGLT1 mutant p.Ala92Val was core-glycosylated but deficient for complex glycosylation, we assumed that the p.Ala92Val mutation caused a defect in the translocation of SGLT1 between the endoplasmic reticulum and the Golgi apparatus. Immunofluorescence microscopy of transiently transfected HEK 293T cells revealed that SGLT1 wild-type and p.Gly43Arg variant were largely detected in the plasma membrane and showed only minor co-staining with the endoplasmic reticulum- and Golgi-specific markers calnexin and giantin, respectively (Figure 5B). The SGLT1 p.Ala92Val mutant, however, showed very distinct intracellular staining and was not present in the plasma membrane. Instead, the p.Ala92Val protein was found to closely match the immunostaining pattern of calnexin, suggesting accumulation in the endoplasmic reticulum of HEK 293T cells.

## 4. Discussion

cGGM represents one disorder within the growing list of congenital diarrheas and enteropathies, which constitute more than 60 rare monogenic diseases [19]. The first symptom of cGGM is watery diarrhea that begins within hours to a few weeks after birth at feeding. Diarrhea is osmotic and causes hypernatremic dehydration with immediate cessation of diarrhea and rapid rehydration upon initiation of a fructose-containing formula [20]. Our cohort of eleven affected individuals presented with clinical manifestations consistent with the diagnosis of cGGM. cGGM has a good prognosis if detected in a timely manner, but delayed diagnosis may result in infantile death [7]. Indeed, three of the four families in our study had children who were deceased in the neonatal period due to diarrhea and most likely represented undiagnosed individuals with cGGM.

Apart from acute and chronic dehydration and metabolic acidosis, inappropriate growth and weight gain were observed in a number of individuals with cGGM [7,9]. Three of the eight affected individuals in our study had short stature, and one patient had both low weight and short stature. The persistence of growth retardation after cessation of diarrhea might be related to malnutrition due to the low socioeconomic status of some families.

cGGM, which causes severe diarrhea in the neonatal period, regresses with age [21], although the speed and extent of recovery vary individually [22], and jejunal glucose transport remains absent. The affected individuals in this study also tolerated carbohydrates as they grew older, in concordance with those previous studies. Thus, a low-carbohydrate diet can be gradually added to the long-term management of individuals with cGGM.

The nonsense variant p.Arg267* identified in families one and three in our study was previously reported in one affected individual with cGGM and nephrocalcinosis (NC) and proximal tubular dysfunction [17]. One of the six affected individuals with the same nonsense variant in this study also had NC. Renal manifestations, including nephrocalcinosis, are common in cGGM, as three of eight affected individuals evaluated for these findings demonstrated NC. Moreover, a recent literature review found NC or renal dysfunction in 20 out of 107 (18.7%) patients [2,7]. Even though the exact mechanism is still unclear for renal manifestations, chronic diarrhea leading to dehydration and concentrated urine was proposed as a possible mechanism, given that *SLC5A1* shows a very weak expression in the kidney. These findings should reinforce the importance of high fluid intake and prevention of dehydration in addition to long-term monitoring of renal status, including regular ultrasound examinations in all affected individuals [9].

The prevalence of cGGM differs in various populations, with an apparent increase in regions with higher rates of consanguinity, such as in Amish and Arabic populations [2,7] and in rural isolates [5]. Two affected individuals with cGGM of Turkish origin were previously reported as case reports in the literature [10,17], and one who carried the p.Arg267* nonsense variant. A founder effect may be hypothesized for the nonsense variant detected in this study, since it was observed in an affected individual of Turkish origin.

The rare *SLC5A1* variants identified in our study were considered to underlie cGGM in affected individuals based on co-segregation with the disease in families, in silico prediction results, molecular modeling, and expression in polarized cells, respectively. The nonsense variant p.Arg267* was predicted to cause nonsense-mediated mRNA decay or a massively truncated SGLT1 protein with six instead of fourteen transmembranes (TM) domains. Missense variants identified in families two and four in this study localized to TM1b and within TM0, respectively [6]. Previously, functional investigation of about 25 missense variants had shown that SGLT1 mutations resulted in protein misfolding, prevented trafficking to the plasma membrane, or were linked to a decreased sugar transport activity. Previously, functional characterization of about 25 missense variants had shown that SGLT1 is variously misfolded and retained in the cell, that SGLT1 is mistransported to the plasma membrane, or that channels with reduced activity are formed [23].

The novel p.Ala92Val variant is retained in the endoplasmic reticulum, as shown by glycosylation analyses and immunofluorescence studies in non-polarized and polarized cells. Structural modeling of p.Ala92Val indicates a reorganization of intramolecular interactions that are likely to influence the three-dimensional organization of SGLT1. Interestingly, both TM1 and TM7 helixes, which are most altered by the mutation, not only constitute the substrate-binding pocket of SGLT1 but also, together with TM13, form the cholesterol-binding site [6]. It has been reported that the presence of cholesterol was required for the efficient endoplasmic reticulum-to-Golgi transport and was consistent with the cholesterol-mediated localization of SGLT1 to lipid rafts [24,25,26]. Furthermore, the mutation of Trp67 in TM1, which makes prominent contacts with a cholesterol moiety, significantly impaired transport and the thermostabilizing effect of cholesterol [6]. The predicted structural changes in TM1 and TM7 of SGLT1 p.Ala92Val may affect the binding of cholesterol and cause the aberrant accumulation of this mutant in the endoplasmic reticulum. The p.Gly43Arg mutant SGLT1 shows processing and localization in polarized CaCo-2 cells comparable to wild-type SGLT1. We hypothesize that the exchange of the highly conserved glycine-43 to arginine impairs the transporter function at the plasma membrane. However, structural modeling was not feasible, as this amino acid and the TM0 helix are not fully represented in the current structure of human SGLT1 [6].

In order to emphasize the potentially pathogenic nature of this missense variant, we performed whole genome sequencing, which did not reveal potentially pathogenic variants in cis. However, we are aware of the fact that this approach does not completely exclude such variants, such as (balanced) structural variants, epimutations, etc. Unfortunately, transcriptomic analyses to at least detect the expression of both alleles and demonstrate normal splicing could not be performed because appropriate patient samples were not available. SGLT1 transport studies are required to ultimately demonstrate the pathogenicity of the p.Gly43Arg variant.

Diagnosis of cGGM is still challenging due to the rarity of the disease, and the clinical picture is difficult to distinguish from other types of CODEs, especially intestinal disaccharidase and transporter deficiencies. Because it is a rare disease, a high clinical suspicion is essential for early detection, and prompt treatment of cGGM is critical to prevent death. Early onset chronic diarrhea in affected individuals should alert pediatricians to consider CODEs. Fasting (nil per os) and targeted molecular studies are the two most important diagnostic methods.

Genetic testing is not strictly necessary to establish the diagnosis, but allows the identification of affected individuals early, and exome sequencing is suited to detect or exclude additional monogenic disorders, thus preventing serious complications and improving clinical outcomes [27]. It is highly recommended to confirm the clinical diagnosis by genetic testing and thus provide a basis for genetic counseling, carrier screening, and future family planning.

Limitations of this study are that the SLC5A1 transcript with the p.Gly43Arg variant could not be analyzed in order to exclude potentially causal variants in cis, and that direct demonstration of abnormal transport function was not performed for this variant. 

## Figures and Tables

**Figure 1 genes-14-01359-f001:**
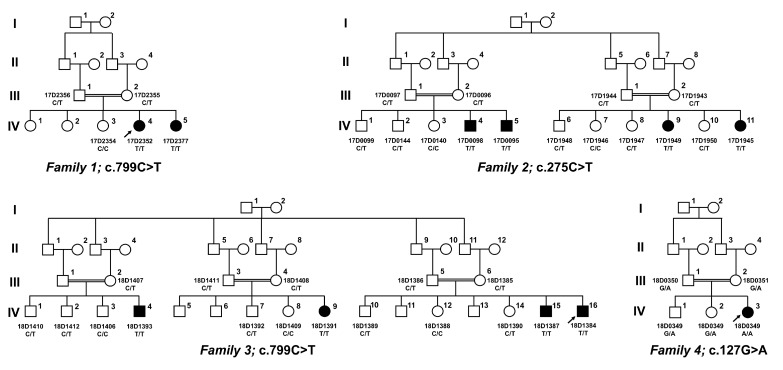
Pedigrees and *SLC5A1* genotypes of the families in this study. Affected individuals are presented by filled symbols, and unaffected individuals by empty symbols. Circles and squares denote females and males, respectively. The genotypes were presented under each individual.

**Figure 2 genes-14-01359-f002:**
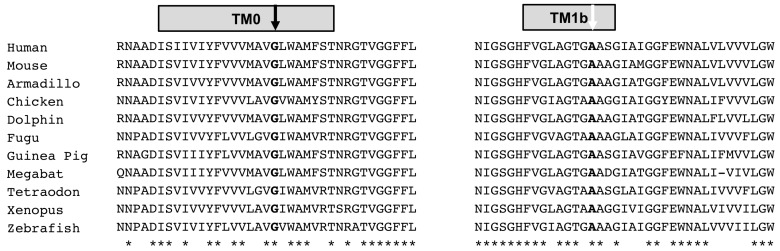
Novel *SLC5A1* missense variants affect highly conserved residues. Evolutionary conservation across species is shown for glycine-43 (black arrow) and alanine-92 (white arrow): TM, transmembrane helix.

**Figure 3 genes-14-01359-f003:**
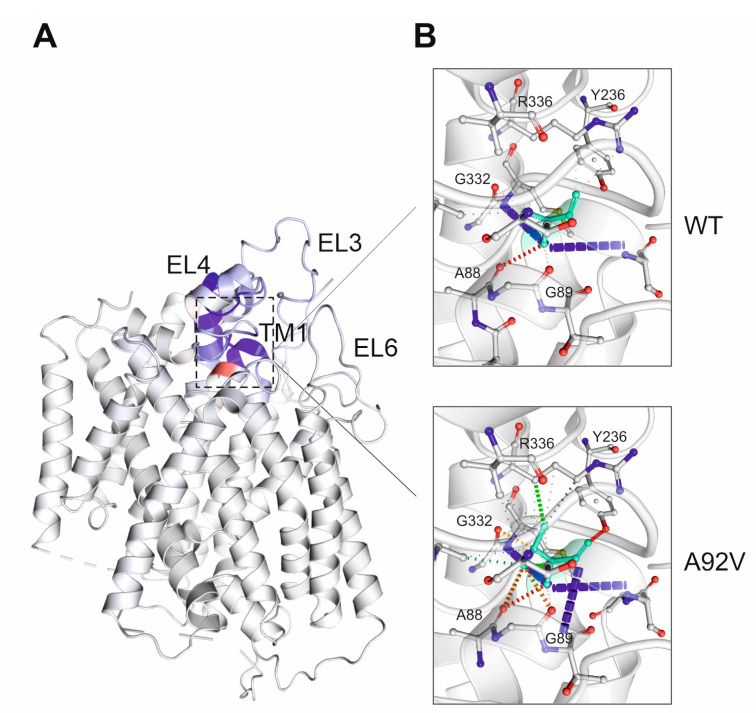
In silico analysis of SGLT1 protein stability and structural changes caused by p.Ala92Val substitution. (**A**) Ribbon representation of the human SGLT1 p.Ala92Val mutant and vibrational entropy changes (ΔΔS_Vib_) between wild-type and mutant; blue regions indicate a rigidification of the structure, and red regions indicate a gain in flexibility. (**B**) Visualization of differences in the non-covalent molecular interactions between wild-type and mutant SGLT1. Ala92 and Val92 residues, respectively, are colored green and are represented as sticks alongside the surrounding residues. The colors are based on the following: red and orange, hydrogen and water-mediated hydrogen bonds; yellow, mixed ionic van der Waals; green, hydrophobic contacts—EL, extracellular loop; TM, transmembrane helix.

**Figure 4 genes-14-01359-f004:**
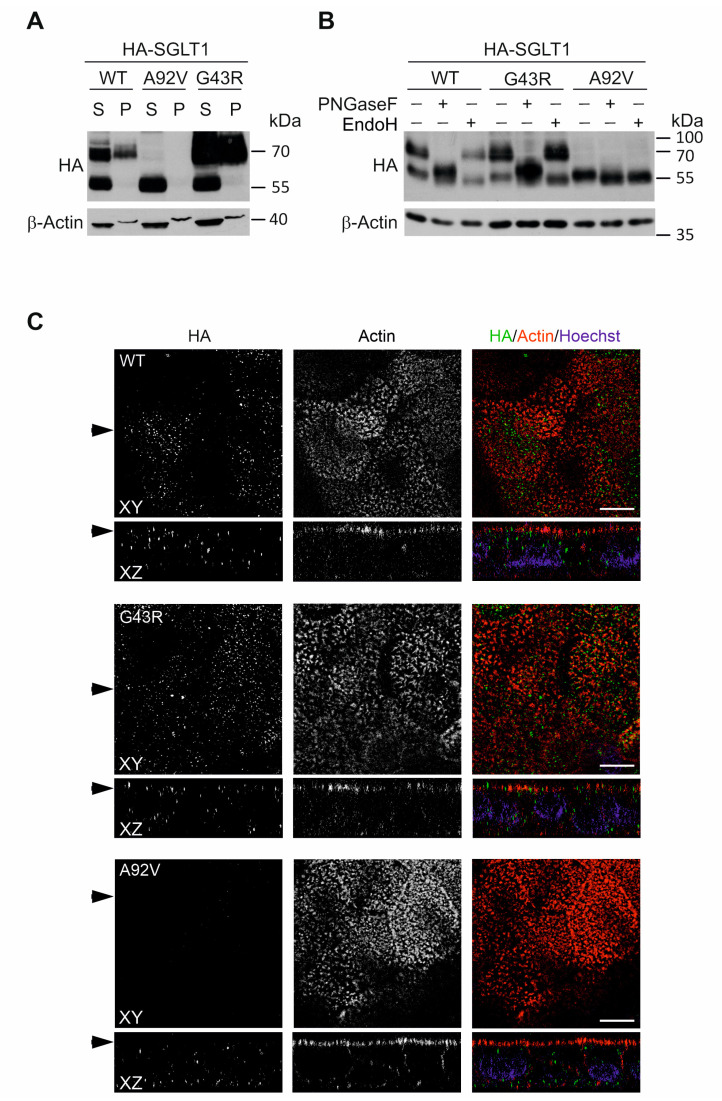
Processing and expression analyses of SGLT1 variants in CaCo-2 cells. (**A**) Immunoblot analysis of ectopic HA-SGLT1 variants expression in NP-40 soluble (S) and insoluble (P) fractions of CaCo-2 cells. β-Actin was used as a loading control. (**B**) Immunoblot analysis of SGLT1 glycosylation in untreated, PNGase F- and Endo H-treated CaCo-2 cell lysates. The lower band at ~60 kDa is present in all NP-40 soluble samples and represents the core glycosylated protein sensitive to PNGase F- and Endo H-specific deglycosylation. The upper bands at ~70 kDa in wild-type SGLT1 and p.Gly43Arg mutant correspond to complex glycosylated proteins and are resistant to Endo H treatment. P.Ala92Val mutant lacks the Endo H resistant glycosylated forms. (**C**) Confocal laser-scanning immunofluorescence images of polarized CaCo-2 cells expressing wild-type HA-SGLT1, p.Gly43Arg or p.Ala92Val variants. Cells were grown for 14 days and stained for HA-SGLT1 and actin—arrowheads on the left mark the corresponding XY and XZ planes. XY represents a single optical section of the apical membrane. The fluorescent signal for pAla92Val was not detected in the apical membrane of polarized CaCo-2 cells: scale bars, 10 µm.

**Figure 5 genes-14-01359-f005:**
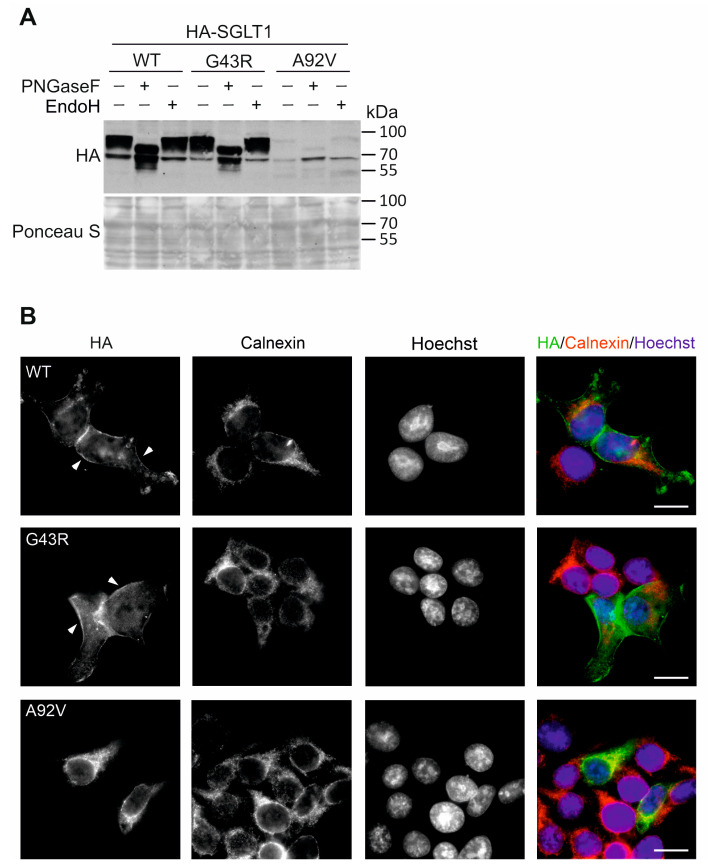
Processing and expression analyses of SGLT1 variants in transiently transfected HEK 293T cells. (**A**) Immunoblot analysis of ectopic expression and glycosylation of wild-type and mutant HA-SGLT1 proteins in HEK 293T cells. Cells transfected with HA-SGLT1 were lysed and incubated with PNGase F- and Endo H, and SGLT1 proteins were detected using anti-HA antibodies. Ponceau S staining of the nitrocellulose membrane was used as a loading control. (**B**) Subcellular localization of HA-SGLT1 (green) in HEK 293T cells was analyzed by epifluorescence microscopy. Immunostaining of calnexin (red) was used to visualize the endoplasmic reticulum: scale bars, 10 µm. Original immunoblot images are provided as Appendix A.

**Table 1 genes-14-01359-t001:** Clinical and laboratory data of eight affected individuals with cGGM. BUN—blood urea nitrogen, SD—standard deviation.

Patient	Gender	Age at Onset (d)	Current Age (y)	Birth Weight (gr)	Weight (kg) (SD)	Height (cm) (SD)	WBC ×10^3^/µL	Hb g/dL	Platelet ×10^3^/µL	BUN mg/dL	Creatinine mg/dL	Na mEq/L	K mEq/L	CI mEq/L	HCO_3_ mmol/L	Stool Reducing Substances (Grade 1 to 4 Positive)	Clinical Assessment of Dehydration
1	F	3	10	3000	23.5 (−1.41)	129 (−0.87)	13.8	12.1	314	49	0.76	151	3.0	129	7.02	+	Severe
2	F	4	7	3200	13.4 (−4.31)	98 (−5.26)	12.6	15.3	239	42	0.85	149	3.2	117	7.17	+	Severe
3	M	15	9	3000	22.5 (−1.64)	128 (−0.75)	14.14	12.9	524	51	0.47	148	2.78	103.6	7.1	+	Moderate
4	M	6	4	3500	14.5 (−0.91)	101 (−0.25)	13.650	8.5	572	39	0.47	147		117	22	+	Moderate
5	M	9	10	3000	29 (−0.44)	132 (−0.42)	12.8	19	260	329	2.06	176	4.37	151	12.2	+	Severe
6	M	11	8	3400	19 (−1.95)	115 (−2.44)	30.64	20.9	194	344	5.37	196	8.13	166	4.3	+	Severe
7	M	3	4	3800	19 (−1.95)	115 (−2.44)	12.92	20.2	353	65	1.42	166	3.87	138	13.4	++++	Severe
8	F	2	3	3060	12 (−0.88)	82 (−2.65)	20.6	19.8	213	38	1.46	186	8.26	144	11.4	+++	Severe

**Table 2 genes-14-01359-t002:** *SLC5A1* variants identified in this study.

Family ID Affected Individual(s)	Variant (*SLC5A1*, NM_000343.3)	Location	Zygosity	GnomAD (v.2.1.1)	Long Contiguous Stretches of Homozygosity
1 2	c.799C>T p.Arg267* +	Exon 8	Homozygous	Not observed in homozygous state **	23.5 Mb on 22q12.3
2 4	c.275C>T p.Ala92Val	Exon 3	Homozygous	Not observed	23 Mb on 22q12.3
3 4	c.799C>T p.Arg267* +	Exon 8	Homozygous	Not observed in homozygous state **	n.d.
4 1	c.127G>A p.Gly43Arg	Exon 1	Homozygous	Not observed	17 Mb on 22q

+ Variants previously reported in the literature in affected individuals with cGGM, ** Observed only in a heterozygous state with an allele count of 4 out of 251,352 (Allele frequency < 0.0001); n.d., not determined.

## Data Availability

Not applicable.

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
