# Peer review of "SLC5A1 Variants in Turkish Patients with Congenital Glucose-Galactose Malabsorption"

_genes, 2023, doi:10.3390/genes14071359_

Round 1

Reviewer 1 Report

Brief Summary

The article by HoÅŸnut et al. studied a group of 11 individuals with congenital glucose-galactose malabsorption (cGGM) from 4 unrelated consanguineous Turkish families. Specifically, the study identified two novel SLC5A1missense variants and further discovered that the p.Ala92Val variant does not translocate to the plasma membrane properly, while the p.Gly43Arg variant reaches the plasma membrane normally but may have impaired glucose transport activity at the plasma membrane. The specific mode of action for the p.Gly43Arg variant to impact regular SGLT1 function remains unclear.

Overall, this study enriched the understanding of cGGM and expanded the spectrum of genetic variants associated with the disease, enabling better detection of the disease in individuals affected in the future.

General Concept Comments

1.     An in vitro transporter function study to confirm reduced glucose transport activity of the two novel variants would greatly enhance the findings. For example, comparing content of intracellular radio-labeled 2-deoxy-D-glucose in cells expressing either WT or one of the novel SGLT1 variants identified in the study in a cell culture system.

Specific Comments

1.     There are currently two Figure 2s in the article. They need to be relabeled.

Author Response

Reviewer 1:

The article by HoÅŸnut et al. studied a group of 11 individuals with congenital glucose-galactose malabsorption (cGGM) from 4 unrelated consanguineous Turkish families. Specifically, the study identified two novel SLC5A1missense variants and further discovered that the p.Ala92Val variant does not translocate to the plasma membrane properly, while the p.Gly43Arg variant reaches the plasma membrane normally but may have impaired glucose transport activity at the plasma membrane. The specific mode of action for the p.Gly43Arg variant to impact regular SGLT1 function remains unclear.

Overall, this study enriched the understanding of cGGM and expanded the spectrum of genetic variants associated with the disease, enabling better detection of the disease in individuals affected in the future.

General Concept Comments

  1. An in vitrotransporter function study to confirm reduced glucose transport activity of the two novel variants would greatly enhance the findings. For example, comparing content of intracellular radio-labeled 2-deoxy-D-glucose in cells expressing either WT or one of the novel SGLT1 variants identified in the study in a cell culture system.

Response:

We appreciate the reviewer's positive feedback on the manuscript.

We also agree that the in vitro sugar uptake studies would be an additional proof of the pathogenicity of SGLT1 variants described in our manuscript. However, such studies require the development of an appropriate experimental cell system that allows the specific and conclusive analysis of SGLT1-mediated sugar transport. We would like to point that CaCo-2 cells, stably transduced with the HA-SGLT1 lentivirus constructs, also express the endogenous SGLT1 protein, which is likely to interfere with the functional analysis of mutants. Although we used the anti-HA-tag antibodies to specifically analyze the expression and subcellular localization of ectopic HA-SGLT1 mutants in these cells, the sugar uptake studies cannot rely on this approach. Furthermore, CaCo-2 cells are known to express a high level of another glucose transporter GLUT2 at both the apical and basal sites. Because 2-deoxy-D-glucose is a non-selective substrate that can be transported across the membrane by both SGLT1 and GLUT2, the substantial compensatory effects are expected. One of the solutions would be to generate SGLT1, GLUT2 or double knockout cell lines followed by reconstitution with ectopic SGLT1 constructs. However, this work would clearly exceed our requirements for a timely revision and we are, therefore, unfortunately unable to carry out this work in this way. Nevertheless, in the present study, we focused primarily on characterizing the clinical and genetic aspects of novel SGLT1 variants that together with our in vitro studies strongly support their pathogenicity.

Specific Comments

  1. There are currently two Figure 2s in the article. They need to be relabeled.

Response:

Thank you, we corrected this error throughout the text.

Reviewer 2 Report

Ferda Özbay HoÅŸnut et. al. investigated the SLC5A1 variants in Turkish patients with congenital glucose-galactose malabsorption. It is an innovative study with some interesting data. Here are comments from the reviewer:

1.     It is weird that the internal controls, beta-actin, of different lanes are of different molecular weights (Figure 3A).

2.     The left and middle pictures should be colored ones, instead of black and white (Figure 3C).

3.     Three figures are not enough for a paper.

4.     The ponceau red staining is not necessary for Figure 1S, because ponceau red staining is suitable for quantification of protein loading. The authors should perform the western blotting of internal control, such as beta-actin, GAPDH.

5.     Please provide the raw uncut film for all western blotting pictures, striped for the incubation of target protein and internal control sequentially.

6.     Also, the fluorescent staining pictures of Figure S1B should be colored ones.

7.     The transfection efficiency is too low according to Figure S1B.

8.     The transfection efficiency of Figure S1B is different from that of Figure 3C.

Author Response

Reviewer 2:

Ferda Özbay HoÅŸnut et. al. investigated the SLC5A1 variants in Turkish patients with congenital glucose-galactose malabsorption. It is an innovative study with some interesting data. Here are comments from the reviewer:

We appreciate the reviewer's thoughts and comments as well as the opportunity provided to address the issues raised.

  1. It is weird that the internal controls, beta-actin, of different lanes are of different molecular weights (Figure 3A).

Response:

We could explain this by technical aspects likely caused by differences in the buffer composition of detergent-soluble and insoluble fractions, e.g. ionic strength, detergent etc. that may affect the apparent mobility of proteins on SDS-PAAG (PMID: 20195791).

  1. The left and middle pictures should be colored ones, instead of black and white (Figure 3C).

Response:

It is best practice to show each channel in its own panel in grayscale when a multichannel image presented in a publication format (PMID: 29953344). The colors are usually introduced by a processing software (please see the "Materials and Methods") based on the use of fluorophores and are shown in a merged image for better visualization.

  1. Three figures are not enough for a paper.

We leave this to the Editorial board.

  1. The ponceau red staining is not necessary for Figure 1S, because ponceau red staining is suitable for quantification of protein loading. The authors should perform the western blotting of internal control, such as beta-actin, GAPDH.

Response:

The experiments carried with transiently transfected HEK293T cells were not controlled by immunoblotting for housekeeping proteins. The loading was assessed by staining the nitrocellulose membranes with Ponceau S. Thus, we would like to use in Figure S1A the Ponceau S-stained membrane as a valid and relevant protein loading control.

  1. Please provide the raw uncut film for all western blotting pictures, striped for the incubation of target protein and internal control sequentially.

Response:

As recommended by the reviewer, we have included original, unadjusted images as supplementary information (Figure S2), this Figure S2 provides an overview. We are prepared to upload all original images at request (size approximately 50 Mb).

  1. Also, the fluorescent staining pictures of Figure S1B should be colored ones.

Response:

Please see our response to the point 2.

  1. The transfection efficiency is too low according to Figure S1B.

 Response:

In the transient transfection experiments, we applied the calcium precipitation method that is widely used in transfecting cells but the efficiency is typically low.

  1. The transfection efficiency of Figure S1B is different from that of Figure 3C.

 Response:

In Figure 3C (revised Figure 4C), we used CaCo-2 cells that were stably transduced with lentiviruses encoding for SGLT1 variants and selected for the expression of ectopic proteins using a selective antibiotic, puromycin. In Figure S1B, however, HEK293T cells were transiently transfected and were not selected for expressing cells, which explains the overall lower efficiency.

Round 2

Reviewer 2 Report

The authors are reluctant to revise the manuscript. 

Author Response

Dear Editors, we replied to all points raised by the Reviewers!

We are happy to address any other points of concern that might arise!

Andreas Janecke